# Fluid Biomarkers of Neuro-Glial Injury in Human Status Epilepticus: A Systematic Review

**DOI:** 10.3390/ijms241512519

**Published:** 2023-08-07

**Authors:** Giada Giovannini, Stefano Meletti

**Affiliations:** 1Neurology Department, Azienda Ospedaliera-Universitaria di Modena, 41126 Modena, Italy; stefano.meletti@unimore.it; 2PhD Program in Clinical and Experimental Medicine, University of Modena and Reggio-Emilia, 41121 Modena, Italy; 3Department of Biomedical, Metabolic and Neural Sciences, University of Modena and Reggio-Emilia, 41121 Modena, Italy

**Keywords:** biomarkers, neuro-glial injury, prognosis, status epilepticus, NSE, TAU, Nf, UCH-L1, S100B, GFAP

## Abstract

As per the latest ILAE definition, status epilepticus (SE) may lead to long-term irreversible consequences, such as neuronal death, neuronal injury, and alterations in neuronal networks. Consequently, there is growing interest in identifying biomarkers that can demonstrate and quantify the extent of neuronal and glial injury. Despite numerous studies conducted on animal models of status epilepticus, which clearly indicate seizure-induced neuronal and glial injury, as well as signs of atrophy and gliosis, evidence in humans remains limited to case reports and small case series. The implications of identifying such biomarkers in clinical practice are significant, including improved prognostic stratification of patients and the early identification of those at high risk of developing irreversible complications. Moreover, the clinical validation of these biomarkers could be crucial in promoting neuroprotective strategies in addition to antiseizure medications. In this study, we present a systematic review of research on biomarkers of neuro-glial injury in patients with status epilepticus.

## 1. Introduction

In recent years, biomarkers have garnered attention for various medical conditions, including different neurological diseases [1]. The latest definition of status epilepticus (SE) by ILAE [2] emphasizes that SE can have long-term consequences (after time point t2), such as neuronal death, neuronal injury, and alterations in neuronal networks, depending on the type and duration of seizures. This definition highlights the concept that sustained seizure activity can lead to additional brain damage, beyond that related to the underlying cause of SE. Potentially irreversible consequences, such as epilepsy development and cognitive and behavioral deficits, could emerge solely as a result of epileptic activity, even in an otherwise healthy brain.

In this context, identifying biomarkers of neuronal and glial injury could serve as a measure of acute seizure-induced brain damage. Therefore, complementing clinical–electroencephalographic evaluation, a multimodal approach based on measuring neuro-glial injury biomarkers could aid in promptly identifying patients at risk of developing short- and long-term consequences of SE.

Based on preclinical evidence, various potential biomarkers of neural and glial degeneration in patients with epilepsy and SE have been proposed so far [3,4,5]. However, currently, none of these biomarkers have been validated for everyday clinical practice.

Preclinical efforts have outlined the primary drivers of epileptogenesis, namely, inflammation, neuron loss, plasticity, and circuit reorganization. Notably, common histopathologic features of epilepsy of diverse origin are microglial activation and astrogliosis, heterotopic neurons in the white matter, loss of neurons, alterations in astrocyte gap junction, and the presence of inflammatory cellular infiltrates [6]. The identification of in vivo epilepsy biomarkers in humans that could track such alterations could be of paramount importance to allow better patient selection, to guide the clinical testing of new hypotheses, and to promote treatment strategies to prevent epileptogenesis.

In this review, we focused on analyzing and summarizing the evidence on biomarkers of neuro-glial injury in humans with status epilepticus. For inflammatory biomarkers, readers are referred to recent reviews on this topic [7,8,9].

## 2. Methods

The results of this systematic review, without meta-analysis, are reported following the recommendations of the Preferred Reporting Items for Systematic Reviews and Meta-Analyses (PRISMA) statement and the Synthesis Without Meta-analysis in systematic reviews (SWiM) extension [10,11].

To identify relevant studies, we conducted a search in MEDLINE (accessed through PubMed as of June 2023, week 2) using the following search terms: “status epilepticus” and “biomarkers”. We included all neuroglial injury biomarkers mentioned in the research strategy, namely, “NSE”, “S100B”, “neurofilament”, “UCHL-1”, “tau”, “*p*-tau”, and “GFAP”. For each biomarker, we first examined evidence from pre-clinical research and then focused on and reviewed all human studies. No date limitations were imposed, and the search was limited to English-language titles and abstracts.

We considered the following types of studies for inclusion: cohorts, case-control, cross-sectional, clinical series, and case reports. Self-reported surveys, reviews/meta-analyses, editorials, commentaries, and expert opinions were excluded.

Participants in the included studies had to be patients presenting with status epilepticus, and they could be of any age, sex, and ethnicity. Two review authors independently assessed the studies for inclusion, and any disagreements were resolved through joint discussion. From the included studies, we extracted the main study author and year of publication, the investigated fluid biomarkers, and the clinical characteristics of the participants. 

## 3. Results

The selection process is presented in Figure 1. We included 27 studies for a total of 738 adult and pediatric patients with SE. 

### 3.1. Fluid Biomarkers of Neuronal Injury in Status Epilepticus

A biomarker is a measurable indicator of a condition or biological process. In recent years, there has been increased interest in brain biomarkers for neurological conditions, including epilepsy and status epilepticus. However, despite a plethora of published research on basic science and clinical research, there are currently no clinically approved brain biomarkers for diagnosing and prognosing epilepsy and SE.

Biomarkers can potentially be measured in various biofluids, but minimally invasive and peripheral biomarkers (such as serum or plasma) play an important role in evaluating brain diseases in humans. An ideal biomarker for SE should meet several criteria: (1) represent molecular and cellular alterations that occur in SE, exist in cerebrospinal fluid (CSF) with higher concentrations in the brain parenchyma than in plasma, and exudate when the blood–brain barrier (BBB) is opened; (2) be stable in body fluids and readily available at the bedside; (3) be easy to assess biochemically and reliable; and (4) demonstrate high sensitivity and specificity, meaning it is low or undetectable under normal conditions [3,5].

When seizures appear and become sustained, they can provoke dysfunction of the BBB with increased permeability. This crucial modification allows leukocytes and albumin to enter and activate microglia and astrocytes, leading to neuroinflammation and oxidative stress, which, in turn, increase neuronal and glial damage. In this environment, neuronal excitability increases, and the seizure threshold decreases, sustaining seizures in a vicious cycle [3,5].

Identifying prognostic biomarkers that predict worse outcomes in patients with SE, including mortality, morbidity, cognitive and behavioral sequelae, and epilepsy development, would be extremely helpful for preventative interventions in high-risk patients [7]. Among these interventions, anti-inflammatory or antioxidant therapies could be added alongside anti-seizure medications, potentially modifying long-term post-SE sequelae. Neuro-glial degeneration biomarkers play a major role among potentially prognostic biomarkers, and they are the focus of this systematic review.

#### 3.1.1. Neuron-Specific Enolase

Neuron-specific enolase (NSE) is one of the first and most extensively studied biomarkers of neuronal injury. NSE is a glycolytic enzyme that catalyzes the conversion of 2-phosphoglycerate to phosphoenolpyruvate. It exists in three dimeric forms: α, β, and γ. The γ fraction is exclusive to neurons and neuroectodermal tissue and, therefore, referred to as NSE. Neuronal enolase is primarily composed of two gamma–gamma subunits, while glial cells contain NSE with two alpha–alpha subunits [3].

NSE is released due to leakage across an injured neuronal membrane, and elevations in serum NSE are indicative of neuronal injury and increased permeability of the blood–brain barrier (BBB). Consequently, it is considered a biomarker of neuronal injury in various neurological conditions, such as stroke, where it correlates with the volume of cerebral infarction [12]. Moreover, in clinical practice, it is used to prognosticate post-anoxic encephalopathy [13].

Regarding SE, numerous preclinical studies on different animal models suggest that NSE levels increase after SE. The elevations in serum NSE are accompanied by tissue evidence of neuronal damage, and the extent of serum NSE elevation appears to be roughly proportional to the degree of irreversible histologic damage [14] and BBB permeability [15].

Most clinical studies on NSE in patients with SE date back to the mid-1990s. Studies of De Giorgio et al. [16,17,18] demonstrated that serum NSE is elevated in the major subtypes of SE (generalized convulsive, complex partial, absence, and subclinical SE), and NSE levels correlated with the outcome and duration of SE. On average, NSE peaked at 41 h after the onset of SE, with 41% of subjects reaching peak levels within 24 h and 33% within 24 to 48 h. Furthermore, the presence of very high NSE levels in “complex partial” SE provided evidence that brain injury occurs even in non-convulsive forms of SE without a co-occurring acute brain insult. Consistent with these results, Rabinowicz et al. [19] reported a significant elevation in serum NSE in two patients with non-convulsive SE compared to patients with epilepsy (sampled far from the last seizure) and healthy controls, suggesting that seizures themselves could cause brain injury [17]. These findings support the need for earlier treatment, even in non-convulsive forms of SE.

In a study by Correale et al. [20], an increase in CSF NSE content was found within 24 h of SE in patients with cryptogenic/remote SE compared to control subjects, along with evidence of altered BBB integrity. Moreover, CSF levels of NSE were significantly higher than serum levels.

In a recent study by Hanin et al. [21], serum NSE levels appeared higher in SE compared to those found in the epilepsy control group, and CSF NSE levels were higher in SE than in healthy controls. However, NSE showed lower discrimination power when compared to two other measured biomarkers (S100B, a marker of astrocytic activation, and progranulin, a marker of neurogenesis/neuronal recovery). In a cohort of 81 SE patients, equal serum levels of NSE and S100B were found in patients who died and those who survived, as well as in those who showed recovery or worsened at hospital discharge. Nevertheless, statistical models containing combinations of different clinical–biochemical variables, including NSE, predicted poor outcomes at discharge and long-term outcomes better than existing clinical scores [4].

Moreover, in eleven patients admitted to the intensive care unit for RSE, who underwent a minimum of three days of continuous EEG monitoring concomitantly with daily serum NSE and S100B assays, the relation of these two biomarkers and EEG scores able to predict the seizure risk were investigated. NSE levels positively correlated with EEG scores, suggesting that NSE could be a potential biomarker of the risk of seizure recurrence. In fact, NSE levels above 17 ng/mL were found to be associated with seizure in 71% of patients, and an increase of more than 15% in NSE levels was associated with seizure recurrence in 80% of patients [22].

In children with SE, findings appear less clear. Wong et al. [23] found that CSF levels of NSE were equal in SE and in single seizures and not different from controls. Elevated levels were found only in those patients presenting symptomatic etiologies. These findings suggest that seizure-induced neuron-specific enolase elevation is rare in children and may occur primarily in patients with severe underlying neurological disorders. In other words, children may be intrinsically more resistant to seizure-induced neuronal injury relative to adults. In the same direction, Shirasaka et al. [24] found that children/adolescents with atypical absence SE (a generalized SE subtype of idiopathic/genetic origin, not associated with structural brain lesions) showed no increase in serum/CSF levels of NSE, while patients with focal onset SE did. On the contrary, O’Regan et al. [25], in a group of 17 children/adolescents with different syndromic epilepsies and etiologies, found that the serum levels of NSE were significantly increased in those with continuous epileptiform discharges (e.g., hypsarrhythmia and electrical status epilepticus) compared to those without and normal controls.

More recently, in the study of Wang et al. [26], an increment in serum levels of NSE, S100B, and vascular endothelial growth factor (VEGF) was found in children within one week from convulsive SE compared to controls. The authors concluded that these biomarkers could serve as a diagnostic marker for SE.

Table 1 reports all NSE findings in human studies.

#### 3.1.2. UCHL-1

Ubiquitin C-terminal hydrolase (UCH-L1) is a cytoplasmic enzyme specific to neurons. Among the three relevant enzymes in this class (UCHL-1, 2, and 3), UCHL-1 is abundantly expressed in neurons, constituting 1–5% of total brain protein. Outside the CNS, UCHL-1 is found in smaller amounts, primarily in ovaries and testes.

The primary function of UCHL-1 is to remove oxidized or increased proteins during normal or pathological conditions by participating in the ubiquitination process. It has been established as a reliable and potential biomarker of neuronal damage, with its CSF levels rising soon after acute neurological insults such as ischemic stroke, subarachnoid hemorrhage, and traumatic brain injury [27,28]. This neuronal protein can be readily detected in CSF and blood very early after injury (within 3 h) and reaches peak levels within 24 h, providing a valuable time-window for potential neuroprotective strategies [29].

Pre-clinical evidence of epilepsy has shown that in kindled rats with intraperitoneal pentylenetetrazol injections, plasma UCHL-1 levels were increased compared to controls, and levels were proportionally elevated in relation to the number of injections [30]. Additionally, the downregulation of UCHL-1 following SE appears deleterious to neuronal survival, interferes with seizure termination, contributes to hyperexcitability, and exacerbates seizure-induced cell death [31].

In human studies, Asadollahi and Simani, Yasak et al., and Elhady et al. [32,33,34] demonstrated elevated serum levels of UCHL-1 in children with epilepsy compared to healthy controls and found a correlation with seizure severity, suggesting the possibility of using UCH-L1 as a biomarker of neuronal damage to monitor disease progression and severity and for early identification of those with drug-resistant epilepsy and those who need epilepsy surgery.

Mondello et al. [35] measured UCH-L1 in 52 patients within 24 h of a single or recurrent epileptic seizure and in 19 healthy controls. UCH-L1 concentrations were significantly higher in patients within 48 h after an epileptic seizure in CSF and within 12 h in serum compared to controls. Moreover, patients with recurrent seizures (four seizures) had significantly higher plasma UCH-L1 concentrations than patients with one or two seizures. Although nine patients presented SE, details on those patients were not reported separately.

Similarly, in a study by Li et al. [36], CSF levels of UCHL-1 appeared to be more elevated in patients with epilepsy than in controls, with higher levels in patients with generalized seizures as well as in those with repetitive seizures, and a correlation with seizure duration and severity was observed too.

Table 2 summarizes the studies on UCH-L1 in human SE.

#### 3.1.3. TAU/p-TAU

TAU protein is a microtubule-associated protein (MAPT) primarily found in neurons in the CNS, where it participates in microtubular assembly and maintains the structure of the axon. Elevated levels of total tau (t-TAU) and phosphorylated tau (*p*-TAU) in CSF indicate both axonal and neuronal damage in many neurological disorders, such as Alzheimer’s disease (AD), traumatic brain injury (TBI), acute ischemic stroke (AIS), viral encephalitis, and Creutzfeldt–Jakob disease (CJD) [1].

In an intra-amygdala KA mouse model of status epilepticus, Alves et al. [37] found an increase in t-TAU mainly located in CA3 and an increase in *p*-TAU in CA1 and CA3 hippocampal subfields, respectively, starting from four hours after KA injection. At the same time, they observed signs of neurodegeneration in the hippocampus ipsilateral to the injection site 24 h after the injection, mainly located in the CA3 hippocampal subfield.

The role of TAU as a biomarker of neuronal and axonal injuries after SE in humans is controversial. Palmio et al. [38] analyzed t-TAU and *p*-TAU levels in 54 patients with single seizures and repeated seizures, including 9 patients with SE. The authors concluded that abnormal CSF tau levels may be found in patients with seizures, but there was no correlation between the presence of SE and t-TAU or *p*-TAU levels. Thus, they concluded that finding increased CSF TAU content in a patient with an epileptic seizure only increases the probability of a symptomatic etiology (i.e., structural brain damage/lesion).

In 2013, Shahim et al. [39] evaluated the role of different biomarkers (t-TAU, NfL, GFAP, and CSF:albumin ratio) in the CSF of children with epilepsy (including four with SE), different neurological conditions, and controls. Overall, among patients with epilepsy, the highest biomarker concentration was found in patients with SE, and in particular, CSF levels of t-TAU were significantly increased compared to those found in patients with focal epilepsy and idiopathic generalized epilepsy.

The same group [40], in 2014, evaluated the CSF levels of different isoforms of β-amyloid (Aβ peptides: Abx-38, Abx-40, Abx-42, and Ab1-42), soluble APP fragments (sAPP-α/β), t-TAU, *p*-TAU, and heart-type fatty acid binding protein (H-FABP) in patients with epilepsy, including four with non-convulsive SE. Reduced CSF levels of Aβ peptides (Aβx38-40-42) were found in NCSE compared to repetitive partial seizures. Overall, reduced levels of t-TAU and *p*-TAU were found in the seizure group compared to controls, while no differences were found in the other biomarkers (Aβ peptides, APP, and H-FABP). Moreover, no differences were found in levels of Aβ1-42, APP, t-TAU, *p*-TAU, and H-FABP within seizure subgroups. The authors concluded that seizures do not have an effect on Aβ metabolism and do not induce axonal damage.

In 2015, our group analyzed t-TAU and *p*-TAU CSF levels in 28 adult patients with SE, excluding cases due to acute symptomatic and progressive causes in order to limit the confounding effect of underlying structural lesions (i.e., stroke and tumors). The t-TAU protein was the most sensitive marker of neuronal damage related to SE [41], and t-TAU levels correlated with the duration of SE, refractoriness of SE to ASM, and with disability development as well. In the same year, Tumani et al. [42] analyzed CSF characteristics (lactate, albumin quotient, cell count, and TAU content) in 309 patients with epileptic seizures, (among which 54 patients had SE) and 10 patients had psychogenic non-epileptic seizures (PNESs). TAU protein levels appeared significantly elevated in patients with CSE followed by NCSE compared to epileptic patients and patients with PNES.

Motojima et al. [43] evaluated the CSF levels of TAU in 33 children with febrile SE, finding that high levels (>500 pg/dL) increased the likelihood of acute encephalopathy, accompanied by the danger of sequelae.

Sood et al. [44] investigated the role of CSF t-TAU in children with SE and found that t-TAU levels were significantly lower in SE patients than in controls, without significant correlations with type, duration, etiology, response to antiseizure medications, poor sensorium, outcome of SE, and critical care needs. Thus, these authors concluded that CSF t-TAU was not a useful diagnostic or prognostic biomarker in pediatric SE.

Table 3 reports all the studies on TAU/*p*-TAU in humans.

#### 3.1.4. Neurofilaments

Neurofilaments (Nfs) are considered biomarkers of axonal damage and injury. They belong to the family of intermediate filaments (IFs) with a diameter of 10 nm, intermediate in size between actin filaments (6 nm) and myosin filaments (15 nm) found in muscle cells. There are six major classes of IF proteins, with Nfs belonging to class IV. Their primary function is to maintain the structure of the axonal cytoskeleton by cross-bridging with other filaments. Nfs are abundant in myelinated axons and are essential for their radial growth, structural stability, and optimal propagation speed of electrical impulses [45].

Although synapses were long thought to be only degradative sites for Nfs transported by axons, recent evidence suggests that a specific pool of Nfs is present in synapses. Thus, they play a different role beyond providing static structural support for axon caliber: they maintain synaptic plasticity and dendritic spines and take an active part in long-term potentiation in the hippocampus. Nfs also form cellular scaffolds for docking and organizing the synaptic vesicles, endosomes, and endoplasmic reticulum [45,46,47].

Structurally, all Nf subunits have a tripartite structure: a conserved central alpha-helical rod region, a short variable head domain at the amino-terminal end, and a tail of highly variable length at the carboxy-terminal end. Nfs are highly phosphorylated proteins in brain tissue, with all subunits phosphorylated at their amino-terminal head domain. The differences in tail length mainly determine the differences between the three major classes of Nfs: neurofilament light (NfL), medium (NfM), and heavy (NfH) chains. The high-molecular-mass subunits (NfH and NfM) are extensively phosphorylated along their carboxy-terminal tail domains, with NfH being the most phosphorylated protein. Phosphorylation confers resistance to protease cleavage and, hence, stability to the Nf cytoskeleton in axons [45,46,47].

Since Nfs are specifically expressed in the axons of neurons when they are damaged or diseased, Nfs may be released into CSF and then pass into the bloodstream, where they can be measured. Thus, they have emerged as promising biomarkers for neurodegeneration, axonal injury, axonal loss, and neuronal death in various neurological disorders, including different forms of Charcot–Marie–Tooth disease (CMT), amyotrophic lateral sclerosis (ALS), familial forms of Parkinson’s disease (PD), atypical parkinsonian syndromes, multiple sclerosis (MS), spinal cord injury, spinal muscular atrophy (SMA), Alzheimer’s disease (AD), frontotemporal dementia (FTD), vascular dementia, and post-anoxic encephalopathy [45,47,48,49,50,51,52,53,54]. Moreover, they have appeared to be increased in some neuropsychiatric conditions, such as bipolar disorders [55] and schizophrenia [46].

In healthy individuals, levels of sNfL exhibit a U-shaped distribution with respect to age. High levels are typically observed in newborns, possibly due to the high rate of neuron turnover during brain development. From there, sNfL levels decrease until late childhood and reach a nadir between the ages of 10 and 15 years. Following this, sNfL levels increase linearly until around the age of 60 years, after which they steadily rise, reflecting neuronal loss associated with normal aging. In contrast, there is currently no conclusive evidence regarding the relationship between sNfL and gender [56].

Pre-clinical studies regarding epilepsy and SE show that in the adult rat hippocampus, degradation of Nf and changes in their phosphorylation occur within days after KA SE induction [57,58], even after brief seizures lasting less than an hour. These findings contrast with those found in the hippocampus of immature rats, where the degradation of Nf does not seem to occur, suggesting that the immature brain exhibits remarkable resistance to neuronal death when exposed to severe, continuous seizures. Moreover, there is a lack of mossy fiber sprouting and neuronal death in KA-treated immature rats [59].

In a recent study using the intrahippocampal kainic acid mouse model of SE, Custers et al. [60] demonstrated higher levels of NfL in both cerebral interstitial fluid and plasma during SE compared to sham-injected mice. Additionally, mice treated with diazepam and ketamine to stop SE showed lower NfL levels than vehicle-treated mice. These findings suggest the potential of NfL as a blood-based fluid biomarker for SE.

In recent years, there has been growing interest in Nf in humans with epilepsy and SE. Rejdak et al. [61] investigated the CSF content of NfH and heat shock protein 70 (HSP-70) within 48 h after seizure cessation in 41 patients (20 with a single seizure, 11 with repetitive seizures, and 10 with SE) without acute symptomatic etiology, along with 18 healthy controls. Patients with SE exhibited significantly higher CSF content of NfH compared to those with single seizures and controls, while those with repetitive seizures also showed higher levels compared to controls. The study indicated that NfH measured in the cerebrospinal fluid can serve as a reliable marker of neuronal damage in patients experiencing repetitive seizures or SE.

However, due to its high molecular weight, NfH is not suitable for measurement in serum. More recent studies have focused on NfL. In a study by Lybeck et al. [62], serum NfL levels (sNfL) were measured at 24–48 and 72 h after cardiac arrest in patients with electrographic status epilepticus and compared with those in patients without SE. The results suggested that electrographic SE is associated with higher levels of serum NfL, indicating potential secondary neuronal injury caused by SE itself. The association with GFAP, a marker of glial injury, appeared less clear.

Our group also investigated the role of serum NfL levels during SE as a prognostic biomarker for short-term outcomes in adults. In a pilot study [63] on 30 SE patients without acute brain lesions, sNfL levels were significantly elevated compared to age and sex-matched healthy controls and patients with epilepsy after single seizures. Moreover, sNfL appeared to be a predictor of 30-day clinical worsening, death, and refractoriness to antiseizure medications in SE patients. This role as a prognostic biomarker was further confirmed in a larger and more heterogeneous cohort of 87 patients [64].

These findings were recently replicated by a German group [65]. Their study also showed significantly elevated sNfL levels in patients with SE compared to healthy controls. Furthermore, higher levels were observed in SE of longer duration, while no differences were evident between convulsive and non-convulsive forms, suggesting that NCSE could induce neuronal damage similarly to convulsive SE. The study also found a strong correlation between serum and CSF content of NfL, confirming its reliability as a biomarker for SE. However, no significant correlation was found between NfL and SE refractoriness and short-term mortality, possibly due to variations in sampling timing.

Regarding the pediatric population, Shahim et al. [39] evaluated the role of different biomarkers in the CSF of children with epilepsy (including four with SE), various neurological conditions, and healthy controls. NfL levels in CSF appeared significantly increased in SE patients compared to those with focal epilepsy, idiopathic generalized epilepsy, and the unspecified epilepsy group.

Table 4 summarizes the studies on Nf and SE in humans.

### 3.2. Fluid Biomarkers of Astroglial Injury in Status Epilepticus

#### 3.2.1. S100B

S100B belongs to the S100 protein family, which currently consists of 25 Ca^2+^-binding proteins with different functions but exhibiting structural similarities. The term S100B refers to a protein identified in the mid-1960s, characterized by its solubility in a 100% saturated solution with ammonium sulfate. It is a homodimer (two β subunits) with a structure composed of two α–helix–loop–helix, and in the loop, there is a binding site for Ca^2+^, although Zn can also bind to this site [66].

S100B is primarily expressed in astrocytes, but it is also found in other glial cell types, such as oligodendrocytes, Schwann cells, ependymal cells, retinal Muller cells, enteric glial cells, and in certain neuron subpopulations. Outside the nervous system, it is mainly expressed in adipocytes [67].

Intracellularly, S100B acts as a calcium sensor protein and plays multiple functions: it transfers signals from second messengers; intervenes in cell proliferation, survival, and differentiation; participates in the regulation of cellular calcium homeostasis and enzyme activities; and even interacts with the cytoskeleton.

Initially, S100B was thought to be a biomarker of glial injury because its detection was interpreted as a consequence of its leakage from damaged cells. However, it is now evident that S100B actively participates in the processes accompanying neural injury. The biological activity of S100B is strictly related to its extracellular concentrations. At low (nanomolar) concentrations present in physiological conditions, it has a neurotrophic effect by promoting neurite extension, modulating the long-term potentiation process, protecting neuron survival, countering neurotoxic insults, and increasing scavenger activity of reactive oxygen species (ROS). At high (micromolar) concentrations, it has a neurotoxic/pro-inflammatory effect by interacting with the RAGE receptor (Receptor for Advanced Glycation Endproducts) and TLR-4 receptor (Toll-Like Receptor-4) present on different cells. This leads to increased production of oxygen radicals, proinflammatory cytokines, cellular adhesion molecules, such as COX2 (Cyclooxygenase-2) release, and glutamate-induced neuronal death, inducing mitochondrial dysfunction and apoptosis.

Therefore, at high concentrations, S100B turns astrocytes into a pro-inflammatory/neurodegenerative phenotype capable of inducing neuronal death and neuroinflammation by both direct and indirect pathways [67]. When the BBB is disrupted, S100B from CSF is released into the serum, where its concentrations are lower.

Interestingly, in animal models with different neurological conditions, the administration of anti-S100B neutralizing antibodies or the suppression of S100B production (by administering arundic acid or using S100B knockout mice models), or the prevention of S100B interaction with transcription factor p53 (by administering pentamidine or TRTK12 peptide), has been shown to improve the condition. Conversely, the administration of S100B or its genetic overexpression has been found to worsen the condition. This suggests that S100B could also be a potential therapeutic target [68,69,70,71].

Clinically, S100B was first found to be elevated in the CSF of multiple sclerosis (MS) patients during acute phases of the disease, while it was low during the stationary phases [69]. Later, its elevation was found in other brain pathologies, such as after acute brain injury (e.g., traumatic brain injuries, strokes) [72], neurodegenerative conditions (e.g., AD, PD, and ALS) [73,74,75], congenital and perinatal disorders (21 chromosome trisomy and asphyxiated newborns) [76], and psychiatric disorders (schizophrenia) [77]. Thus, S100B is recognized as a sensitive but not very specific biomarker of active astrocyte distress [66,67]. Similar to neurofilaments, the expression of S100B can be affected by aging, even in healthy controls. Some authors have suggested that the expression of S100B in the brain cortex and hippocampus increases with age due to the activation of astrocytes [66,67].

Regarding status epilepticus, increased levels of S100B in the CSF have been found in a lithium–pilocarpine rat model of SE, which correlated with evidence of neurodegeneration in the dentate gyrus and CA1 hippocampal subfield and an increase in mossy fiber sprouting 24 h after SE induction [78].

In chronic epilepsy, several studies have reported increased serum levels of S100B in epilepsy patients compared to healthy controls, and its concentrations may reflect epilepsy severity, suggesting its potential prognostic value [5,66,79,80,81].

Fewer studies have focused on the role of S100B in SE in humans. An observational study on children with SE found a significant difference in the mean value of S100B between the SE group and the control group (single simple febrile seizure) and a strong positive correlation between S100B levels and the degree of encephalopathy defined by the degree of acute alterations detected by MRI in SE patients. Thus, the authors concluded that S100B levels can help define which acute MRI alterations could be irreversible [82].

In a more recent study [26], significantly increased serum levels of NSE, S100B, and VEGF were found within one week in children with convulsive SE compared to controls, leading the authors to conclude that S100B could serve as a diagnostic marker. In a study by Hanin et al. [21], serum S100B levels were found to be higher in SE patients than in those with epilepsy and healthy controls, while no significant differences were found for CSF S100B levels. No differences were found in relation to SE refractoriness to treatment, semiology, or etiology. Using a cutoff of 0.09 ng/mL, the diagnostic value of serum S100B levels for SE was found to be good (AUC, 0.748; sensitivity, 59.7%; specificity, 90.2%; positive predictive value, 83.6%; negative predictive value, 72.8%). However, in a cohort of 81 SE patients, equal serum levels of NSE and S100B were found in deceased and surviving patients and in those who recovered or worsened at hospital discharge [4]. Additionally, S100B did not correlate with EEG scores that measure the risk of seizure development applied to EEG monitoring in 11 patients after RSE [22].

More recently, our group explored the prognostic role of S100B in a population of 87 adult patients with SE [64]. Serum levels of S100B were overall increased in patients with SE compared to both patients with epilepsy and healthy controls. Even though increased serum levels were found in patients with clinical worsening at 30 days after SE, the level of S100B did not appear as an independent predictor of clinical worsening.

Table 5 reports all the studies on S100B in SE in humans.

#### 3.2.2. GFAP

Glial fibrillary acidic protein (GFAP) is the principal intermediate filament of astrocytes, belonging to class III of intermediate filaments. GFAP is considered a biomarker of astroglial proliferation and reactive astrogliosis.

Animal models have shown that GFAP expression is altered as a result of SE, and signs of gliosis have been observed in different studies. In a study on a mouse model of pilocarpine-induced SE, do Nascimento et al. [83] found that the molecular CA3 and CA1 pyramidal cell layers of the hippocampus expressed the highest presence of GFAP immunoreaction at one and three weeks after SE onset. In rat models of electrically induced non-convulsive SE, Avdic et al. [84] observed excessive microglial and astrocytic activation in the hippocampus at one week after SE, which continued at four weeks accompanied by neuronal loss.

However, SE could also induce early astrocyte degeneration. In a study by Borges et al. [85] in a mouse model of pilocarpine-induced SE, immunoreactivity for GFAP was decreased in the dentate hilus, and the number of healthy-appearing GFAP- or S100B-positive cells was significantly reduced by at least 65% one and three days after SE induction. However, ten days after SE, hilar GFAP immunoreactivity reappeared, indicating astrocyte degeneration followed by astrogenesis, leading to hilar repopulation. Alterations in GFAP expression were not limited to the hippocampus; in a pilocarpine rat model of SE, Schmidt-Kastner et al. [86] demonstrated that three days after SE, immunohistochemistry for GFAP was severely reduced both in the substantia nigra pars reticulata and in basal cortical areas. A reduction in GFAP-positive astrocytes in the hippocampus of rat pups experiencing prolonged febrile seizures was also linked to a reduction in the magnitude of long-term potentiation, leading to impairment of learning and memory in the developing brain [87].

Regarding studies on epilepsy in humans, a transient but marked increase in GFAP was evidenced immediately after (2 h) a single tonic–clonic (TC) seizure [88]. GFAP increase was more prominent than that of TAU, NfL, and UCHL-1, indicating possibly greater involvement of glial cells than neurons. Moreover, elevated GFAP levels in children with epilepsy were found to be a predictor for active seizures, and it was suggested that they could be used to monitor disease progression and severity for early identification of those with drug-resistant epilepsy and those who are in need of epilepsy surgery [34]. In line with these results, a recent study by Akel et al. [89] in which increased levels of GFAP and NfL were found in patients with recent seizures in a large cohort of patients with epilepsy.

Fewer studies have focused on the role of GFAP in humans with SE. Gurnett et al. [90] showed that CSF levels of GFAP were significantly increased within 24 h after a single seizure in children compared to controls. Seizure duration was found to be positively correlated with GFAP levels, and levels measured in patients with SE were significantly higher compared to those achieved in patients without SE. In these patients, no differences in NSE were found, suggesting that GFAP may be a more sensitive marker of brain injury than NSE. This result was confirmed in a study by Wang et al. on a cohort of children with convulsive SE, in which serum levels of GFAP were found to be significantly increased on the fourth day after SE compared to healthy controls [26].

On the contrary, in a cohort of adult patients with post-cardiac arrest electrographic status epilepticus, even though GFAP levels were significantly increased at 72 h, electrographic status epilepticus did not appear as an independent predictor of serum GFAP levels (while it was for NfL). The authors concluded that the association of SE with GFAP appeared less clear, and they suggested that the short half-life (48 h) of GFAP may have played an important role in the results [62].

Table 6 reports all the studies on GFAP in SE in humans.

## 4. Discussion

Animal models of status epilepticus clearly show neuronal injury and glial activation, accompanied by pathological evidence of neuronal loss and gliosis induced by persistent continuous ictal activity that exacerbates the initial injury that triggered seizures [14,78,91].

In humans, studies on biomarkers of neural and glial injury in status epilepticus overall confirm that both biomarkers of neural/axonal injury and glial activation are increased during and immediately after SE, with differences related to the specific single biomarker.

However, it is important to emphasize that the data obtained in preclinical animal models are mainly derived from the observation of alterations in different biomarkers on tissue and not on biological fluids, with some exceptions [30,60,78]. In this sense, on the one hand, it is important to be cautious in formulating parallels between humans and data in the animal. On the other hand, the current possibility of researching and assaying molecules/biomarkers on fluids in humans makes it attractive to develop animal studies that evaluate these same biomarkers to better guide clinical research in humans.

Among these molecules, NSE shows the greatest number of studies, and its elevation in both central and peripheral body fluids appears in adult patients during or shortly after SE, while its role is less clear in the pediatric population. On the contrary, evidence is less robust for TAU, where the two largest studies in adults show concordant results demonstrating increased levels during SE, while in children, evidence is scarce and more contrasting. Studies on the role of UCH-L1 in humans are extremely limited.

In recent years, NfL has gained increased attention both in pre-clinical and clinical studies, and it appears to be a sensitive biomarker of axonal injury during SE and a predictor of a more severe form of SE. An important advantage of NfL is its reliability in measurement in serum/blood. Additionally, two studies on humans [63,65] and a preclinical study [60] have demonstrated a close and optimal correlation between the peripheral and central compartments.

As for biomarkers of astroglial activation, even though studies show an increment in both S100B and GFAP in SE patients, the relationship with the outcome appears less strong than for biomarkers of neural injury.

Overall, and particularly for NSE and TAU, the reviewed papers documented relatively poor reproducibility. In our opinion, the reasons for the variability in the observed results depend on several factors, which are important, especially for future studies, to try to control. The first factor is inherent in the heterogeneity of the condition studied (SE). For this very reason, precise phenotyping of the case series that is adherent to the current ILAE classification in terms of etiologies, semeiology, age, and EEG patterns is essential. From the analysis of the included papers, it is evident, particularly for papers prior to 2015, how outdated terminologies and clinical classifications were used.

A second crucial factor concerns the time of sampling: in many studies, it is not given whether biomarkers were dosed when the patient was still in SE, or after SE resolution, and in these cases, at what distance from seizure resolution. It is very likely that this variable may largely influence the single biomarker result. Therefore, in future studies, it will be crucial to standardize and make explicit the time of sampling with respect to the onset and resolution of SE.

Finally, few studies addressed the potential roles of several confounding factors such as renal function, body mass index, and comorbid conditions, like heart failure or diabetes.

For these reasons, human studies are limited in number and have important limitations that leave several aspects undetermined and many questions still unanswered.

To this point, we focused our attention on the biomarkers of neuronal and astroglial injury, which were already investigated in biofluid in human SE. However, several other potential biomarkers would be worthy of investigation. For example, the expression of gap-junction-alteration-related proteins, such as connexin 43 and connexin 32 [92,93], and the expression of activity-regulated cytoskeleton-associated proteins could provide information on the expression of biomarkers of epileptogenesis already documented in animal models [94].

## 5. Conclusions and Future Perspectives

The results observed in humans overall support the idea that repeated seizures in SE result in increased biomarkers of neuronal and astroglial damage. However, in order to improve knowledge and clarify the many aspects that are still obscure to date, some improvement actions are needed for future studies.

Firstly, studies on humans consist primarily of case reports or small case series, with a lack of large multicentric prospective cohorts. Therefore, the current results are difficult to generalize, and it is not possible to clearly understand how these biomarkers behave in specific subpopulations of SE patients (e.g., in relation to different etiologies, clinical semiology, age at SE onset, response to therapy, etc.).

Secondly, only a few studies consider repeated samples at different time points in the same patient, making it difficult to understand the temporal variation of these biomarkers in both acute and chronic phases. Moreover, while we know that biomarkers of neural and glial injury are elevated in both CSF and serum during and immediately after SE, the situation during chronic phases is not well understood. Even though there is evidence of a strict interplay between neurodegenerative diseases and epilepsy with seizures accelerating the progression and exacerbating outcomes of neurodegenerative diseases [95], it remains unclear whether the persistence of continuous ictal activity leads only to acute injury or if mechanisms of self-sustained neurodegeneration occur [96,97,98]. Moreover, reports of gliosis and atrophy following SE in humans are mainly derived from isolated case reports or small case series [99,100,101], where multiple factors other than the persistence of epileptic activity often contribute to their appearance.

Finally, studies evaluating the role of biomarkers in predicting long-term outcomes in terms of both functional and cognitive outcomes and epilepsy development are completely lacking. The identification of biomarkers able to identify, from the outset, which patients are at risk of developing irreversible damage would allow potential neuroprotective strategies to be applied and monitored over time [102].

For the future, and to arrive at the identification of one or more biomarkers that can be validated in clinical practice, it is necessary to develop prospective, multicenter studies with the aim of both including large case series representative of the great heterogeneity of SE and performing *longitudinal* assessments in the acute and chronic phases after the cessation of SE. To achieve this goal, the use of biobanks and the development of shared procedures for the collection of biological material and its preservation and analysis are equally important steps [103].

## Figures and Tables

**Figure 1 ijms-24-12519-f001:**
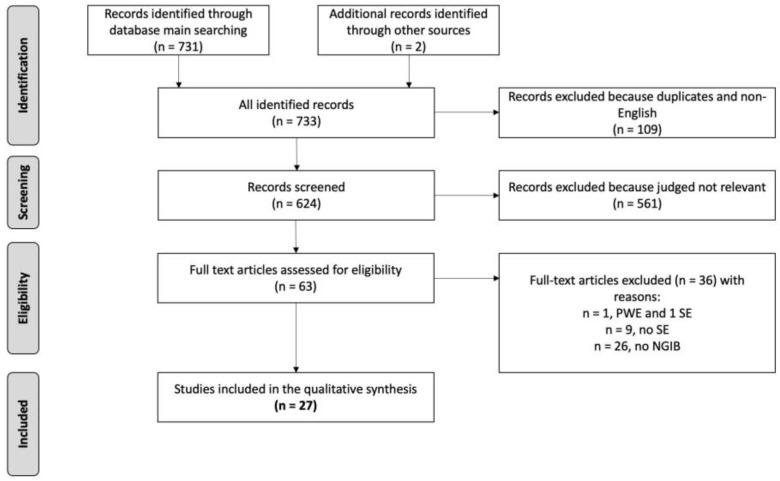
Selection process flow chart. PWE: patients with epilepsy; SE: status epilepticus; NGIB: neuro-glial injury biomarkers.

**Table 1 ijms-24-12519-t001:** Studies on NSE in humans with status epilepticus.

Author/Year	N of SE	Age; Gender	Biomarkers	Types of SE	Body Fluids	Sampling during SE	Principal Findings
De Giorgio et al., 1995 [16]	19 (vs. PWE + HC)	NR	NSE	NR	Serum	N	↑ NSE in SE vs. PWE and HC
Rabinowicz et al., 1995 [19]	2 (vs. 13 PWE + HC)	38, 40 y; 2F	NSE	NCSE	Serum	N	↑ NSE in NCSE
De Giorgio et al., 1996 [17]	8 (vs. 18 HC + 13 PWE)	NR	NSE	CPSE (non-lesional)	Serum	NR	↑ NSE in CPSE vs. PWE and HC
Correale et al., 1998 [20]	11 (vs. HC)	Mean age, 42.2 ± 13.6 y; 6M, 5F	NSE	NR (cryptogenic and remote symptomatic)	CSF and serum	NR	↑ CSF-NSE in SE vs. HC
O’Regan et al., 1998 [25]	Ped: 17 (with continuous ED), 16 (w/o ED) + HC	Mean age, 7.3 y; range, 5 m-17 y; 9M, 8F	NSE	NCSE	Serum	Y	↑NSE in PWE with continuous ED vs. PWE w/o and vs. HC
De Giorgio et al., 1999 [18]	31 (vs. 13 PWE, + 30 HC)	NR	NSE	GCSE, ASE, CPSE, NCSE in coma	Serum	NR	↑ NSE in SE vs. HC. ↑ NSE in CPSE and NCSE in coma ↑ NSE and longer SE
Shirasaka et al., 2002 [24]	Ped: 4 (2 vs. 2)	2-8-11-18 y; 2M, 2F	NSE	NCSE (ASE vs. CPSE)	CSF and serum	N	Normal levels of NSE in ASE
Wong et al., 2002 [23]	Ped: 19 (vs. 30 with SS + 4 with brain tumors + 39 HC)	PWE: mean age, 55.2 ± 62.8 m; range, 4–212 m; gender, NR	NSE	CSE and NCSE	CSF	NR	NSE levels equal in SE and SS. NSE elevated in symptomatic etiologies
Hanin et al., 2022 [21]	82 (vs. 36 HC + 56 PWE)	Mean, 49.8 ± 18.8 y; ratio M/F: 60	NSE, PRG, S100B	CSE and NCSE	CSF and serum	Y	↑ s-NSE, progranulin, and S100B in SE vs. PWE and HC. No differences in relation to SE refractoriness, semeiology, and etiology
Hanin et al., 2022 [22]	11	Mean, 41.0 ± 20.6 y; range, 20–75 y; gender, NR	NSE, S100B	CSE and NCSE RSE	Serum	N	NSE as a biomarker of EEG activity and of risk of seizure recurrence
Hanin et al., 2022 [4]	81	Mean, 50 ± 19 y; 49M, 32F	NSE, S100B, PRG, and a panel of lipid and biochemistry *	CSE and NCSE	CSF and serum	Y	= NSE and S100B in dead and alive, in recovered and worsened Models with different variables among which NSE predict poor outcome at discharge and long-term outcome better than existing clinical scores
Wang et al., 2022 [26]	Ped 57 (vs. 30 HC)	Range, 1–133 m; 32M, 25F	NSE, S100B, VEGF, GFAP, CRP	CSE	Serum	Y	↑ NSE, S100B, VEGF, GFAP, CRP

* Neuron-specific enolase (NSE), S100B, progranulin, sodium, potassium, chloride, urea, creatinine, aspartate aminotransferase, alanine aminotransferase, gamma gt, lactates, bilirubin, hemoglobin, platelet count, white blood cell count, neutrophil/lymphocyte ratio, total cholesterol (TC), triglycerides, HDL-cholesterol (HDL-C), LDL-cholesterol (LDL-C), TC/HDL-C, apolipoprotein A1 (ApoA1), apolipoprotein B (ApoB), ApoA1/HDL-C, ApoB/LDL-C, lipoprotein(a), apolipoprotein E, lipoprotein-associated phospholipase A2, free cholesterol, esterified cholesterol (EC), cholesterol esterification ratio (EC/TC), phospholipid (PL), TC/PL. F: female; M: male; y: years; m: months; PWE: patients with epilepsy; HC: healthy control; NR: not reported; ↑ increment; = equal; N: no; Y: yes; NCSE: non-convulsive status epilepticus; CPSE: complex partial status epilepticus; ED: epileptiform discharges; GCSE: generalized convulsive status epilepticus; ASE: absence status epilepticus; SS: single seizure; CSE: convulsive status epilepticus; RSE: refractory status epilepticus; PRG: progranulin, CRP: C-reactive protein; VEGF: vascular endothelial growth factor; GFAP: glial fibrillary acid protein.

**Table 2 ijms-24-12519-t002:** Studies on UCH-L1 in humans with status epilepticus.

Author/Year	N of SE	Age; Gender	Biomarkers	Types of SE	Body Fluids	Sampling during SE	Principal Findings
Mondello et al., 2012 [35]	9 (vs. 27 SS, 16 RS, 19 HC)	PWE: mean: 45.5 y, range: 31.5–62 y; 30M, 22F	UCH-L1	NR	Serum and CSF	NR	↑ CSF UCH-L1 within 48 h from seizures ↑ Serum UCH-L1 within 12 h from seizures Higher levels in RS (analysis not reported for SE)

F: female; M: male; y: years; m: months; SS: single seizure; RS: repetitive seizure; HC: healthy control; NR: not reported; ↑ increment.

**Table 3 ijms-24-12519-t003:** Studies on TAU/*p*-TAU in humans with status epilepticus.

Author/Year	N of SE	Age; Gender	Biomarkers	Types of SE	Body Fluids	Sampling during SE	Principal Findings
Palmio et al., 2009 [38]	9 (vs. 45 SS or RS)	PWE: mean, 48 y; range, 16–88 y; 30M, 24F	t-TAU, *p*-TAU	NR	CSF	NR	No increased levels of t-TAU, *p*-TAU in SE. Increased levels of TAU in patients with seizures of symptomatic cause
Shahim et al., 2013 [39]	Ped: 4 (vs. 113 PWE, 79 HC, 411 OND)	PWE: median, 1.4 y; range, 0–16; 56M, 44F	t-TAU, NfL, GFAP, CSF:albumin ratio	NR	CSF	NR	↑ t-TAU in SE compared to PE and PGE
Shahim et al., 2014 [40]	4 (vs. 25 SS, 16 RS, 17 HC)	Median, 39 y; range, 30–56 y; 3M, 1F	Aβ peptides, APP, t-TAU, *p*-TAU, H-FABP	NCSE	CSF	N	↓ Aβ peptides (Aβx38-40-42) in NCSE compared to RPS; Overall ↓ t-TAU, *p*-TAU and =Aβ peptides, APP and H-FABP in seizures compared to controls; = levels of Aβ1-42, APP, t-TAU, *p*-TAU, H-FABP within seizure subgroups
Monti et al., 2015 [41]	28	Mean, 56 y; range, 11–79 y; 10M, 18F	t-TAU, *p*-TAU, Aβ	CSE and NCSE	CSF	N	↑ t-TAU in RSE/SRSE, longer SE, worsening of clinical conditions and epilepsy development
Tumani et al., 2015 [42]	54 (vs. 255 PWE, 10 PNES)	PWE: mean, 55 y; range, 14–97 y; 170M, 159F	TAU, cell count, lactate, albumin quotient	CSE and NCSE	CSF	NR	↑ TAU in SE compared to PWE and PNES
Motojima et al., 2016 [43]	Ped: 33 (13 FC, 20 w Enc)	FC: mean, 3 y 8 m; 7M, 6F. Enc: mean, 2 y 11 m; 9M, 11F	CSF: t- TAU and IL6; Serum: AST, ALT, LDH, NH3, INR	NR	CSF and serum	NR	↑ t-TAU in patients developing sequelae
Sood et al., 2021 [44]	Ped: 50 (vs. 15 HC)	Range, 6 m-12 y; 31M, 19F	t-TAU	CSE	CSF	NR	↓ t-TAU in SE vs. HC. No correlation with type, duration, etiology, response to treatment, outcomes, level of consciousness, ICU admission

SS: single seizure; RS: repeated seizure; F: female; M: male; y: years; m: months; NR: not reported; N: no; NCSE: non-convulsive status epilepticus; HC: healthy control; OND: other neurological disorder; GFAP: glial fibrillary acid protein, NfL: neurofilament light chain; APP: amyloid-beta precursor protein; H-FABP: heart-type fatty acid binding protein; ↑ increment; ↓ decrement; = equal; RPS: repeated partial seizures; PE: partial epilepsy; PGE: primary generalized epilepsy; CSE: convulsive status epilepticus; RSE: refractory status epilepticus; SRSE: super-refractory status epilepticus; PNES: psychogenic non-epileptic seizure; FC: febrile convulsion; Enc: encephalopathy; AST: aspartate aminotransferase; ALT: alanine transaminase; INR: international normalized ratio; IL6: interleukine 6; ICU: intensive care unit.

**Table 4 ijms-24-12519-t004:** Studies on neurofilaments in humans with status epilepticus.

Author/Year	N of SE	Age; Gender	Biomarkers	Types of SE	Body Fluids	Sampling during SE	Principal Findings
Rejdak et al., 2012 [61]	10 (vs. 20 SS, 11 RS, 18 HC)	Median, 52 y; range, 29–71 y; 6M, 4F	NfH and HSP-70	CSE	CSF	N	↑ NfH in SE vs. SS and HC ↑ NfH in worse outcome
Shahim et al., 2013 [39]	Ped: 4 (vs. 113 PWE, 79 HC, 411 OND)	PWE: median, 1.4 y; range, 0–16; 56M, 44F	t-TAU, NfL, GFAP, CSF:albumin ratio	NR	CSF	NR	↑ NfL in SE compared to PE, PGE, UE
Lybeck et al., 2021 [62]	26 (vs. 102 non-ESE)	Median, 72 y; IQR, 65–81; 21M, 5F	NfL and GFAP	SE in post-anoxic encephalopathy (ESE)	Serum	Y	↑ s-NfL in ESE vs. non-ESE.
Giovannini et al., 2022 [63]	30 (vs. 30 PWE + 30 HC)	Mean, 45 ± 19.9 y; range, 11–79 y; 16M, 14F	NfL	CSE and NCSE (w/o acute symptomatic)	Serum and CSF (17)	Y	↑ s-NfL in SE (vs. PWE and HC) ↑ s-NfL in RSE/SRSE, SE > 24 h S-NfL predictor of 30 d worsening or death
Giovannini et al., 2023 [64]	87 (vs. 30 PWE + 27 HC)	Median, 70 y; IQR, 25; 33M, 54F	NfL and S100B	CSE and NCSE	Serum	Y	↑ s-NfL in SE (vs. PWE and HC) s-NfL predictor of 30 d worsening
Margraf et al., 2023 [65]	28 (vs. 1186 HC)	Mean, 69.4 ± 15 y; 4M, 24F	NfL	CSE and NCSE	Serum and CSF	N	↑ s-NfL in SE vs. HC ↑ s-NfL in SE of longer duration; equal increment in CSE and NCSE No increment in dead and refractory patients

SS: single seizure; RS: repeated seizure; HC: healthy control; F: female; M: male; y: years; m: months; NfH: neurofilament heavy chain; HSP-70: heat shock protein 70; GFAP: glial fibrillary acid protein; PE: partial epilepsy; PGE: primary generalized epilepsy; UE: unspecified epilepsy; CSE: convulsive status epilepticus; N: no; Y: yes; ↑ increment; ESE: electrographic status epilepticus; NfL: neurofilament light chain; GFAP: glial fibrillary acid protein; PWE: patients with epilepsy; NCSE: non-convulsive status epilepticus; RSE: refractory status epilepticus; SRSE: super-refractory status epilepticus.

**Table 5 ijms-24-12519-t005:** Studies on S100B in humans with status epilepticus.

Author/Year	N of SE	Age; Gender	Biomarkers	Types of SE	Body Fluids	Sampling during SE	Principal Findings
Gunawan et al., 2019 [82]	24 (vs. 22 FS)	Median, 1.5 ± 4.51 y; 16M, 8F	S100B	CSE and NCSE	Serum	NR	↑ S100B in SE (vs. FS) Strong positive correlation with the MRI degree of encephalopathy
Wang et al., 2022 [26]	Ped: 57 (vs. 30 HC)	Range, 1–133 m; 32M, 25F	S100B, NSE, VEGF, GFAP, CRP	CSE	Serum	Y	↑ S100B in SE (vs. HC)
Hanin et al., 2022 [21]	82 (vs. 56 PWE + 36 HC)	Mean, 49.8 ± 18.8 y; ratio M/F: 60	S100B, NSE, PRG	CSE and NCSE	Serum and CSF	Y	↑ s-S100B in SE (vs. PWE and HC). No differences in CSF levels No differences in relation to SE refractoriness, semeiology, and etiology
Hanin et al., 2022 [22]	11	Mean, 41.0 ± 20.6 y; range, 20–75 y; gender, NR	NSE, S100B	CSE and NCSE RSE	Serum	N	S100B is not a biomarker of EEG activity and of risk of seizure recurrence
Hanin et al., 2022 [4]	81	Mean age, 50 ± 19 y; 49M, 32F	NSE, S100B, PRG, and a panel of lipid and biochemistry *	CSE and NCSE	CSF and serum	Y	= NSE and S100B in dead and alive, in recovered and worsened patients
Giovannini et al., 2023 [64]	87 (vs. 30 PWE + 27 HC)	Median, 70 y; IQR, 25; 33M, 54F	NfL and S100B	CSE and NCSE	Serum	Y	↑ s-S100B in SE (vs. PWE and HC) S100B is not a predictor of 30 d worsening

* NSE (neuron-specific enolase), S100B, progranulin, sodium, potassium, chloride, urea, creatinine, aspartate aminotransferase, alanine aminotransferase, gamma gt, lactates, bilirubin, hemoglobin, platelet count, white blood cell count, neutrophil/lymphocyte ratio, total cholesterol (TC), triglycerides, HDL-cholesterol (HDL-C), LDL-cholesterol (LDL-C), TC/HDL-C, apolipoprotein A1 (ApoA1), apolipoprotein B (ApoB), ApoA1/HDL-C, ApoB/LDL-C, lipoprotein(a), apolipoprotein E, lipoprotein-associated phospholipase A2, free cholesterol, esterified cholesterol (EC), cholesterol esterification ratio (EC/TC), phospholipid (PL), TC/PL. FS: febrile seizure; F: female; M: male; y: years; m: months; CSE: convulsive status epilepticus; NCSE: non-convulsive status epilepticus; NR: not reported; ↑ increment; = equal; HC: healthy control; VEGF: vascular endothelial growth factor; GFAP: glial fibrillary acid protein; CRP: C-reactive protein; Y: yes; N: No; PWE: patients with epilepsy; PRG: progranulin; RSE: refractory status epilepticus; NfL: neurofilament light chain.

**Table 6 ijms-24-12519-t006:** Studies on GFAP in humans with status epilepticus.

Author/ Year	N of SE	Age; Gender	Biomarkers	Types of SE	Body Fluids	Sampling during SE	Principal Findings
Gurnett et al., 2003 [90]	Ped: 52 PWE (vs. 33 HC). Num of SE NR	PWE: mean, 47 ± 57 m; range, 5–212 m; gender NR	GFAP and NSE	NR	CSF	N	↑ GFAP in SE compared to non-SE
Lybeck et al., 2021 [62]	26 (vs. 102 non-ESE)	Median, 72 y; IQR, 65–81; 21M, 5F	NfL and GFAP	SE in post-anoxic encephalopathy (ESE)	Serum	Y	↑ GFAP at 72 h, ESE not an independent predictor of GFAP levels
Wang et al., 2022 [26]	Ped 57 (vs. 30 HC)	Range, 1–133 m; 32M, 25F	NSE, S100B, VEGF, GFAP, CRP	CSE	Serum	Y	↑ NSE, S100B, VEGF, GFAP, CRP

PWE: patients with epilepsy; HC: healthy controls; Num: number; F: female; M: male; y: years; m: months; NSE: neuron-specific enolase; NR: not reported; N: no; Y: yes; ↑ increment; ESE: electrographic status epilepticus; NfL: neurofilament light chain; VEGF: vascular endothelial growth factor; CRP: C-reactive protein; CSE: convulsive status epilepticus.

## Data Availability

Not applicable.

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
