# Peer review of "Fluid Biomarkers of Neuro-Glial Injury in Human Status Epilepticus: A Systematic Review"

_ijms, 2023, doi:10.3390/ijms241512519_

Round 1

Reviewer 1 Report

It is very good review. I have only one comment: When you are speaking about pediatric patients the age of patients shoud be presented (if accessible).

The review is easy to read.

Author Response

We thank the reviewer for the positive judgment on our work.

Following her/his suggestion we added in the tables the age and gender of the studied populations when available.

Reviewer 2 Report

The review by Giovannini et al have provided useful insights into attempts in finding relevant biomarkers for seizures in a variety population of patients with epilepsy. Specifically, the review heavily elaborates on the potential fluid biomarkers (both CSF and serum) associated with neuronal damage and glial activation following SE and/or multiple seizure episodes based on current literature findings. This review is extremely valuable and will help readers better understand the potential use of fluid biomarkers to guide the usage of treatments in clinical practice in patients with SE. However, the following comments may be useful to improve the quality of the manuscript.  

While the authors focus more on CSF findings in humans and cite preclinical studies as to why authors look at these biomarkers in human, the animal studies (e.g line 228) reported in this review mostly only evaluate brain pathology rather than CSF finding. Including animal studies that directly investigate these biomarkers in CSF or serum would help to establish a stronger association with human studies.  

H. Tumani et al, 2015 (DOI: 10.1016/j.eplepsyres.2015.04.004) also examined the relationships between status epilepticus (convulsive and non-convulsive) and CSF composition, especially tau levels. However, this study is not included in the review. Is there a reason why this study is excluded?  Given rather inclusive findings on tau levels in CSF of patients with SE/epilepsy, a more thorough discussion of this biomarker is needed to explain why such discrepancies exist.

Although authors acknowledge the caveats of the studies included in this review, they seemed to inadequately address the fundamental issue as to why there is lack of reproducibility in some of the biomarkers studied by different groups and provide potential recommendations to overcome this. The discussion on this matter should be included as it poses a major problem for potential publication bias.   

Author Response

First of all, we thank the reviewer for the overall positive evaluation of our work and for the relevant suggestions and insights to improve the manuscript.

Regarding the first comment the reviewer is right. Unfortunately, there is very little work in animal models that has investigated, in addition to tissue pathology, any biomarkers on biological fluids. Therefore, the work on preclinical models predominantly refers to tissue data. An exception, for example, is the recent work published by Custers et al., (2023 Epilepsia) on neurofilaments in the Kainic acid model of SE. In fact, work such as this one is of particular interest because when substantial convergence is observed between data on animal models and data in humans, the significance and the "robustness" of the inferences that can be drawn is much higher. Probably, until a few years ago since no real investigation of biomarkers on biological fluids in epilepsy, and particularly in status epilepticus had yet begun, there was also little interest in developing this type of analysis on animal models. Instead, it is now evident how "parallel" data on a given biomarker, in humans and in animals, are extremely useful.
We added a specific comment/paragraph in the discussion section on this point acknowledging that this is as a limitation of the study as well as a topic that is important for future investigations.

As far as the study by Tumani et al (2015) we apologize, and we have added this study to the manuscript. The main findings of this study were added to Table 3, and in the text of the paragraph 3.3. TAU/p-TAU.

Finally, we substantially expended the Discussion section to evaluate in more details the “lack of reproducibility in some of the biomarkers studied by different groups and provide potential recommendations to overcome this”. We thank the reviewer for this comment, and we agree that lot of work is needed to overcome actual discrepancy and limitations of published studies.

Reviewer 3 Report

Review of Manuscript “Fluid biomarkers of neuro-glial injury in human status epilepticus: a systematic review” submitted to the International Journal of Molecular Sciences by Giada Giovannini and Stefano Meletti.

This is an interesting review focused on the role of some biomarkers of neuro-glial injury in human status epilepticus (SE). This is a new topic with increased interest and still limited data in the field of epilepsy. The manuscript has presented clinical and some experimental evidence of different potential biomarkers for neural and glial degeneration in patients with epilepsy and SE although at present none of these markers have been validated so can be successfully used in everyday clinical practice. The advantage of biomarkers is that they can be potentially measured easily in various biofluids so that they can play an important role in evaluating brain diseases in humans. The authors have presented in detail the following fluid biomarkers: Neuron-specific enolase, Ubiquitin C-terminal hydrolase, TAU protein, and Neurofilaments. They present neuronal injury during status epilepticus. Furthermore, the authors have presented and the fluid biomarkers of astroglia injury in status epilepticus: S100B which belongs to the S100 protein family and Glial fibrillary acidic protein.

All tables very well summarize, the presented clinical and experimental data.

My specific recommendations are listed below:

1.      There is a mistake in the numeration from 3.3 moved to 4.4.

2.      English spelling check is absolutely required, there are mistakes throughout the text.

Author Response

We thank the reviewer for the positive evaluation of our work.

We corrected the mistake in the numeration of paragraph 4.4. into 3.4.

We also checked through the manuscript for English language editing and mistakes.

Reviewer 4 Report

At the manuscript “Fluid biomarkers of neuro-glial injury in human status epilepticus: a systematic review” by Drs. Giada Giovannini and Stefano Meletti authors reviewed the studies on biomarkers of neuro-glial injury in patients with Status Epilepticus. Without a doubt, the identification of biomarkers of neuronal and glial injury could represent a measure of acute seizure-induced brain damage, but many details of the biological mechanism of this injury remains unclear.

The gap-junction-coupled astroglial network plays a central role in the regulation in the epilepstogenesis. This, experimental models point to a complete loss of astrocytic coupling, but preservation of the gap junction forming proteins connexin-43 and connexin-30. There are a number of studies on the role of gap junctions and possibly related biomarkers. I think it should at least briefly mention this, it is directly related to this manuscript. I think the following publications should be quoted:

Peter Bedner and Christian Steinhäuser; Role of Impaired Astrocyte Gap Junction Coupling in Epileptogenesis; Cells;  2023 20;12(12):1669. doi: 10.3390/cells12121669.

Volnova et al; The Anti-Epileptic Effects of Carbenoxolone In Vitro and In Vivo;  Int J Mol Sci; 2022 8;23(2):663. doi: 10.3390/ijms23020663.

Aronica et al,  Expression of connexin 43 and connexin 32 gap-junction proteins in epilepsy-associated brain tumors and in the perilesional epileptic cortex; Acta Neuropathol 2001;101(5):449-59. doi: 10.1007/s004010000305.

Klein et al; Commonalities in epileptogenic processes from different acute brain insults: Do they translate? Epilepsia . 2018 Jan;59(1):37-66. doi: 10.1111/epi.13965. Epub 2017 Dec 15.

There are very interesting data of recent years on the role of activity-regulated cytoskeleton-associated protein (ARC) in epileptogenesis. The use of Arc as a biomerker is also possible. I would use the following publications and add a few sentences about Arc to the manuscript:

Gol et al; Assessment of genes involved in behavior, learning, memory, and synaptic plasticity following status epilepticus in rats. Epilepsy Behav. 2019 Sep;98(Pt A):101-109. doi: 10.1016/j.yebeh.2019.06.023.

Sibarov et al; Arc protein, a remnant of ancient retrovirus, forms virus-like particles, which are abundantly generated by neurons during epileptic seizures, and affects epileptic susceptibility in rodent models; Front. Neurol., 2023 ; Sec. Epilepsy; V 14  https://doi.org/10.3389/fneur.2023.1201104

Borges et al; Intermittency properties in a temporal lobe epilepsy model. Epilepsy Behav. 139:109072. doi: 10.1016/j.yebeh.2022.109072.  PMID: 36652897

The presentation of a subject is systematic and comprehensive and analysis is proper. I am happy to recommend the manuscript for the publication after minor corrections mentioned above.

Author Response

We deeply thank the reviewer for the positive judgment of our manuscript and for her/his insightful suggestion on two specific topic that we did not address: namely the role of Gap junction (GJ) and of the activity-regulated cytoskeleton-associated protein (ARC) in epileptogenesis.

Indeed, we were focused on biomarkers of neuronal injury and therefore we have not initially considered these two relevant fields of investigation. Following the reviewer suggestion, we have now briefly considered in the Discussion section the relevance for the future to expand biomarkers investigation in human also to GJ and ARC. We also quoted in the introduction the article of Klein et al (2018), while some of the other suggested studies we cited in the new paragraphs of the discussion.

Round 2

Reviewer 2 Report

I thank the authors for addressing all the comments and suggestions from the previous review. Now that all concerns have been resolved, I believe that the manuscript has been strengthened with these additions and modifications, therefore ready for publication.